# Effect of H_2_S Corrosion on the Fracture Toughness of the X80 Pipeline Steel Welded Joint

**DOI:** 10.3390/ma15134458

**Published:** 2022-06-24

**Authors:** Xueli Wang, Dongpo Wang, Caiyan Deng, Chengning Li

**Affiliations:** 1Key Laboratory of Advanced Joining Technology of Tianjin, Department of Materials Science and Engineering, Tianjin University, Road Weijin 92, Tianjin 300072, China; wangxl@pipechina.com.cn (X.W.); dengcy@tju.edu.cn (C.D.); licn@tju.edu.cn (C.L.); 2Pipe China North Pipeline Company, Road Xinkai 408, Langfang 065000, China

**Keywords:** hydrogen sulfide, pipeline steel, fracture toughness, coarse grain heat-affected zone, subcritical grain heat-affected zone

## Abstract

To analyze the causes and mechanisms affecting the fracture toughness of X80 pipeline steel welded joints against H_2_S, the fracture toughness of different zones of X80 pipeline steel welded joints in both air and saturated H_2_S solution was investigated. The fracture toughness of welded joints degraded significantly in the saturated H_2_S solution, where the crack tip opening displacement (CTOD) characteristic value in the coarse grain heat-affected zone (CGHAZ) and weld metal (WM) was only 8% and 12% of that in air, respectively. However, the sub-critical grain heat-affected zone (SCHAZ) showed better resistance to H_2_S corrosion, with the CTOD characteristic value reaching 42% of that in air. The resistance of the welded joint to H_2_S corrosion was sensitive to microstructures. The grain boundary ferrite (GBF) presented in WM, and the angle of grain boundary orientation in CGHAZ was not conducive to hindering crack propagation. Moreover, the formation of the resultant hydrogen cracks owing to the H_2_S corrosion also reduced the fracture toughness of the welded joint.

## 1. Introduction

The service safety of high-grade pipeline steel-welded joints has become one of the important issues that need to be addressed owing to the occurrence of fracture failure accidents, with the rising demand for oil and gas resources [1,2,3,4,5,6,7]. The fracture toughness of welded joints was significantly influenced by the microstructure and service environment. The CGHAZ near the fusion line (FL) experiences excessive growth of grain size owing to a longer period of high-temperature welding thermal cycling (about 1200 °C) in the welding process [8]. The high residual stress is simultaneously introduced to the welded joints owing to the welding thermal cycle [9,10,11,12]. On the other hand, pipeline steel is inevitably exposed to corrosive medium (H_2_S), thus facing problems such as H_2_S/CO_2_ corrosion, electrochemical corrosion, hydrogen embrittlement (HE), hydrogen-induced cracking (HIC), and sulfide stress corrosion cracking (SSCC) [3,13,14,15,16]. Yuxin Chen et al. [17] investigated the influence of H_2_S interaction with prestrain on the mechanical properties of high-strength X80 steel and found that fractography exhibited brittle fracture for H_2_S-introduced specimens and necking phenomenon decreased significantly compared with H_2_S-free specimens. Lijun Gan et al. [18] investigated hydrogen trapping and hydrogen-induced cracking of welded X100 pipeline steel in H_2_S environments and the results showed that the welded joint with an inhomogeneous microstructure had a higher trap density and was more susceptible to HIC due to being two orders of magnitude larger in the concentration of irreversible hydrogen than that of the base metal, though all presented poor HIC resistance for both the base metal and the welded joint. Dejun Kong et al. [13] investigated stress corrosion of X80 pipeline steel welded joints by slow strain test in NACE H_2_S solutions and the results showed that the sensibility index of SCC in NACE solution (saturated H_2_S) was 56.94%, and the plastic loss was the most serious, an obvious stress corrosion tendency appearing. Wei Zhao et al. [19] studied the corrosion behavior of reheated CGHAZ of X80 pipeline steel in H_2_S-containing environments and found that the intercritically reheated CGHAZ had the lowest corrosion resistance because of the coarse necklace-shaped martensite–austenite (M/A) constituents. Yonghe Yang et al. [7] investigated the fracture toughness of the materials in welded joint of X80 pipeline steel and the results showed that the fusion zone (FZ) was the fracture risk zone of the X80 steel weldment owing to the occurrence of hard-brittle (M/A) constituents. Alan Tribe et al. [20] studied the fracture toughness of friction stir-welded API X80 and found that fracture toughness of the weld metal increased linearly with decreases in heat input. The present research focused on the H_2_S corrosion behaviors and fracture toughness of the pipeline steel-welded joints. However, the influence of H_2_S corrosion on the fracture toughness of high-strength pipeline steel-welded joints is an aspect that has been rarely reported.

In this paper, the fracture toughness of X80 pipeline steel-welded joints in saturated H_2_S solution were studied. Gas metal arc welding (GMAW) and Lincoln Pipeliner 80Ni1 welding wire were used for the X80 joints weld. The evolution of microstructure in different zones of the joints was analyzed. CTOD samples of the different zones was prepared and the CTOD experiments were carried out in hydrogen sulfide environment. Further, the causes of fracture toughness deterioration in X80 pipeline steel weld joints in H_2_S-containing environments were revealed.

## 2. Experimental Procedures

### 2.1. Materials and Welding

X80 pipeline steel (base material, BM) with a diameter of 1422 mm was used in this study. The GMAW with the position automatic pipeline welder A610 under the shielding gas of 80% Ar and 20% CO_2_ was employed for the welding process. Lincoln Pipeliner 80Ni1 welding wire with a diameter of 1.0 mm was applied and the wire feeding speed was 400 mm/min during the welding process. The welding voltage and current were 22 V and 140 A, respectively. The welding groove structure and the cross-section of the weld joints are shown in Figure 1. The macroscopic morphology of the weld metal (WM) showed that the layer distribution is uniform. The welding heat input led to the formation of CGHAZ adjacent to the FL. The fine grain heat-affected zone (FGHAZ), inter-critical heat-affected zone (ICHAZ), and sub-critical heat-affected zone (SCHAZ) transitioned from the WM to the BM in sequence. The SCHAZ was 5 mm from the FL.

### 2.2. Experimental Methods

CTOD test was carried out at the WM center, CGHAZ, and SGHAZ. The test operation and sample geometry were conducted according to BSEN ISO-15653 (2018), ISO-12135, and API-1104 (2018). The sampling location is shown in Figure 2; B × B (B = W = 17 mm) type specimens with a length of 120 mm (≥4.6 W) were used. The notch type of the WZ was NP (N: normal to weld direction, P: parallel to weld direction), while the notch type of the CGHAZ and SGHAZ specimens was NQ (N: normal to weld direction, Q: weld thickness direction). The lengths of the mechanical gap and pre-cracks were 4.50 mm and 4.00 mm, respectively. Total crack length was 0.5 times the sample width, which ranged from 0.45 W to 0.7 W. Fatigue pre-cracking was conducted on GPS200 high-frequency fatigue tester. CTOD test was conducted on the electronic universal testing machine which was equipped with a hydrogen sulfide environment chamber, as shown in Figure 3. A CTOD extensometer was used to measure the crack tip opening displacement. The specimens with pre-fabricated fatigue cracks were soaked in saturated H_2_S solution for one week before the CTOD test, and the corrosive environment was maintained by passing H_2_S into the hydrogen sulfide environment chamber during the CTOD test. Micro-hardness test was conducted in a Vickers hardness tester at an indentation load = 98 N and dwell time = 15 s to analyze the variation in hardness of the welded joint. 

The samples were cut across the cross-section of the welded joint, and etched in 4% nitric acid alcohol solution. Microstructural features were examined using the optical microscope (OM), and the JEOL JSM-7800 scanning electron microscope (SEM) equipped with an energy dispersive spectroscopy (EDS) analyzer and electron backscatter diffraction (EBSD). Besides, an accelerating voltage of 20 kV, working distance of 15 mm, and a step size of 0.15 μm were exerted to attain orientation maps via EBSD.

## 3. Results and Discussion

### 3.1. Microstructure of X80 Welded Joint

The microstructure of X80 welded joint at different locations is shown in Figure 4. The WM was dominated by large columnar grains composed of staggered acicular ferrite (AF), grain boundary ferrite (GBF), and a small amount of bainite as shown in Figure 4a,d. The CGHAZ directly affects the overall performance of the welded joint. Figure 4b,e exhibit that the microstructure of CGHAZ was mainly coarse polygonal ferrite (PF) and granular bainite (GB), as well as some lath bainite (LB) and M/A constituents. Figure 4c,f depict that SCHAZ underwent a tempering-like process affected by welding heat input, mainly containing fine PF and AF.

### 3.2. Micro-Hardness

Figure 5 shows the hardness distributions of the X80 welded joint, the hardness of WM was about 254 HV, and the microstructure of WZ was mainly AF (Figure 4a,d). The FGHAZ had the highest hardness of 260 HV owing to its grain refinement. The hardness of CGHAZ was about 257 HV, which was slightly lower than that of the FGHAZ. The hardness of SCHAZ was about 215 HV, which was slightly lower than that of the BM. From the FGHAZ to SCHAZ, the hardness decreased obviously.

### 3.3. Results of CTOD and Micro-Hardness Tests

The loading curve of the CTOD test in air and saturated H_2_S solution are shown in Figure 6, and the results of CTOD characteristic values are shown in Figure 7. The CTOD characteristic values *δ*_0_ are obtained from Equations (1) and (2):(1)$$\delta_{0}={\left[\left(\frac{S}{W}\right)\frac{F}{{\left(BB_{N}W\right)}^{0.5}}\times {\left(\frac{a_{0}}{W}\right)}^{2}\right]}^{2}\left[\frac{\left(1-V^{2}\right)}{mR_{p0.2}E}\right]+\tau \cdot C_{V_{p}}\frac{0.43\left(W-a_{0}\right)V_{p}}{0.43\left(W-a_{0}\right)+a_{0}},$$
(2)$$\mathrm{where}\;C_{V_{p}}=-1.74{\left\{\left(\frac{a_{0}}{W}\right)-0.45\right\}}^{2}+1$$

The symbols in Equations (1) and (2) can be found in the BSEN ISO-15653 (2018) and ISO-12135. Under both test conditions, the fracture toughness of WM was lower than that of the CGHAZ and SCHAZ. In addition, the fracture toughness of the welded joint under the influence of H_2_S was significantly lower compared with the CTOD test results under air environment condition.

From the CTOD test results (Figure 7), the CTOD characteristic values of the WM, CGHAZ, and SCHAZ in the air environment were 0.25 mm, 0.73 mm, and 0.76 mm, respectively. However, the CTOD characteristic values of the WZ, CGHAZ, and SGHAZ in the H_2_S-containing environment were about 0.03 mm, 0.06 mm, and 0.32 mm, respectively. The fracture toughness of the welded joint at different locations in the H_2_S-containing environment was significantly reduced. The CGHAZ and SCHAZ had almost equal CTOD characteristic values in the air environment. However, in the H_2_S-containing environment, the fracture toughness of the WM and CGHAZ degraded significantly and the CTOD characteristic value of WZ and CGHAZ was only 12% and 8% of that in air, respectively. The SCHAZ showed a better resistance to H_2_S corrosion, and the CTOD characteristic value of SCHAZ in the H_2_S-containing environment was 42% of that in air.

### 3.4. Crack Propagation Behaviors of the Joint in H_2_S and Air

Fracture toughness can be measured with the energy required for crack initiation and propagation. In principle, the improvements in fracture toughness of welded joint can be presented in the crack behaviors, such as crack blunting, crack deviation, and crack stoppage. Therefore, to fully understand the effect of H_2_S corrosion on the fracture toughness of welded joint at different locations, the characteristics of crack propagation on CTOD specimens were observed.

Figure 8 shows propagation paths of the main cracks observed from the typical CTOD profiles in the H_2_S-containing environment. The crack propagation path in WZ was quite smooth as shown in Figure 8a, indicating the low crack propagation resistance of WZ under the H_2_S-containing environment. Figure 8b,c show that the crack deviation occurred in CGHAZ and SCHAZ, which suggests that propagation of the cracks in CGHAZ, especially in SCHAZ, needs to consume more energy.

Figure 9 shows the propagation behaviors of secondary cracks in the H_2_S-containing environment and air environment. The cracks at the WM were inclined to propagate within the coarse GBF and be arrested on AF for both testing environments (Figure 9a,d). The WM in the H_2_S-containing environment showed more secondary cracks within GBF compared with the WM in an air environment. The cracks at the CGHAZ tended to propagate inside the PF (Figure 9b,e). The cracks at the CGHAZ tested in the H_2_S-containing environment were arrested on the phase boundaries, while the cracks at the CGHAZ tested in an air environment were arrested on the AF. The cracks at the SCHAZ tended to propagate along the rolling strip and be arrested at the zones of grain refinement.

Figure 10 shows the KAM maps around the propagation paths of secondary cracks with respect to the WM, CGHAZ, and SCHAZ in air, as well as in the H_2_S-containing environment. For WZ and CGHAZ, as shown in Figure 10a,b,d,e, the propensity of the plasticity around the secondary crack path was highest immediately adjacent to the the crack-plane, gradually decreasing with an expansion in distance from the crack under both environment conditions. Moreover, in the presence of H_2_S, the dramatic reduction of crack-wake plasticity was prominent for WZ and CGHAZ, with this difference reflected on the KAM maps. This suggests that the crack propagated without any intense plasticity expansion in the H_2_S-containing environment for WZ and CGHAZ. On the contrary, Figure 10c,f show that the plastic strain distributions in SCHAZ were intense and homogeneous, indicating that the crack propagation in SCHAZ underwent intense plasticity expansion under both environment condition.

The fracture models for hydrogen-assisted crack propagation have been well-established in by experimental and analytical evidence [21,22]. The well-known hydrogen-enhanced localized plasticity (HELP) model, whereby hydrogen locally accelerates dislocation emission and motion, causes the crack to propagate via an extremely localized ductile fracture accompanying the narrower extension of the plastic zone. Therefore, compared with SCHAZ, the WZ and CGHAZ were more sensitive to HE in the H_2_S-containing environment.

### 3.5. Fracture Toughness Deterioration of the Joint in H_2_S Containing Environment

Crack propagation behaviors of the welded joint in the H_2_S-containing environment were significantly influenced by the H_2_S corrosion during loading. In the H_2_S-containing environment, the welded joint would react with the S element and the reduction in fracture toughness of the welded joint is thought to be associated with the hydrogen embrittlement phenomenon. Hydrogen atoms in H_2_S diffuse into the welded joint in the form of protons, and these hydrogen atoms tend to accumulate at locations such as grain boundaries and oxides [23]. Mousavi Anijdan et al. [23] studied the hydrogen cracking phenomenon in API X65 pipeline steels in an H_2_S-containing environment, and found that H_2_S reacts with the material in the following ways (Equations (3)–(6)), leading to the occurrence of hydrogen embrittlement.
(3)$$H_{2}S\to 2H^{+}+S^{2-},$$
(4)$$\mathrm{Fe}+2H^{+}\to {\mathrm{Fe}}^{2+}+2H\;,$$
(5)$${\mathrm{Fe}}^{2+}+S^{2-}\to \mathrm{FeS}$$
(6)$$H_{2}S+\;\mathrm{Fe}\to \mathrm{FeS}+2H,$$

As shown in Figure 11 and Figure 12, The enrichment of S was found near the crack tip. During the CTOD test, H_2_S continuously penetrated through the pre-fabricated fatigue crack toward the crack tip, which makes it easier for H_2_S to accumulate at grain boundaries, dislocations, and inclusions around the cracks, resulting in a large enrichment of hydrogen atoms at these defect locations and the formation of HE. This mechanism is more inclined to take place under applied loads, causing damage of materials [24].

Since the presence of hydrogen is difficult to detect, the enrichment of S in the vicinity of the crack (Figure 12) reflects that hydrogen atoms do play a role in the propagation of the cracks, which was supposed to induce the occurrence of HE and lead to a reduction in the fracture toughness of the welded joint in the H_2_S-containing environment. Wang et al. [25] investigated the corrosion resistance of X80 pipeline steel submerged arc-welded joints to H_2_S in terms of electrochemical corrosion tests and the results showed that the FGHAZ dominated by PF had the best H_2_S corrosion resistance, followed by the WM composed of fine AF, while the CGHAZ consisted of GF and some M-A constituents had the worst H_2_S corrosion resistance. Moreover, it was reported that the bainite in the CGHAZ had the lowest open-circuit potential and it was most susceptible to being corroded during the H_2_S electrochemical corrosion process [25]. The occurrence of GB and LB was thought to cause CGHAZ to exhibit the weakest resistance to H_2_S corrosion. As mentioned above (Figure 11), the CGHAZ generated corrosion products with many holes in the corrosion process, which facilitates the cracks propagating along the direction of external forces and further reduced the crack propagation resistance for CGHAZ. The WZ exhibited a significantly deteriorative fracture toughness in the H_2_S-containing environment due to the presence of coarse GBF. The coarse GBF has lower yield strength and higher subjected plastic strain, so the secondary cracks tend to propagate in the GBF. Besides, the crack propagation resistance of WZ further deteriorated due to the effect of H_2_S corrosion, resulting in the continuous decrease of fracture toughness in the H_2_S-containing environment. 

Figure 13 shows the distributions of grain orientation angles in different locations of the welded joint. The number of low angle grain boundaries (≤10°) was the highest in the CGHAZ, and low angle grain boundaries are related to substructures of dislocations. The increase in low angle grain boundaries facilitates the improvement of strength and toughness [8]. However, grain boundaries larger than 15° have better crack propagation resistance [26,27]. Many scholars have found that high angle grain boundaries in the microstructure could effectively impede crack propagation [15,26]. The results of the grain orientation angle distribution in different regions derived from EBSD revealed that the CGHAZ had the lowest number of high angle grain boundaries, which to some extent reduced the ability of this zone to hinder crack propagation. In addition, as shown in Figure 14, severe coarsening of grain sizes in CGHAZ and the presence of inhomogeneous M/A constituents and GF also deteriorated the fracture toughness of the CGHAZ. The SCHAZ also has a large number of low angle grain boundaries. However, the microstructure of the SCHAZ was homogeneous ferrite grains compared with the WM and CGHAZ, and there were no coarse GBF and M/A constituents. The SCHAZ still maintained the rolling direction of the BM, although it had been influenced by the welding heat input, making it exhibit the inhomogeneity of crack propagation resistance. Therefore, The SCHAZ displayed the best resistance to H_2_S corrosion and crack propagation.

## 4. Conclusions


The CTOD characteristic values of the WM, CGHAZ, and SCHAZ in air were 0.25 mm, 0.73 mm, and 0.76 mm, respectively. However, the CTOD characteristic values of the WZ, CGHAZ, and SGHAZ in the H_2_S-containing environment were about 0.03 mm, 0.06 mm, and 0.32 mm. The fracture toughness of the WZ and CGHAZ degraded significantly in the saturated H_2_S solution, where the CTOD characteristic value in the CGHAZ and WZ was only 8% and 12% of that in air, respectively. The SGHAZ showed better resistance to H_2_S corrosion, with the CTOD characteristic value reaching 42% of that in air.The H_2_S continuously penetrated through the pre-fabricated fatigue crack toward the crack tip, which made H_2_S accumulate at the grain boundaries, dislocations, and inclusions around the cracks, thus leading to a large enrichment of hydrogen atoms at these defect locations and the formation of hydrogen embrittlement. This caused the WZ and CGHAZ to exhibit the dramatic reduction of crack-wake plasticity in the H_2_S-containing environment.The cracks tended to propagate within GBF for WZ due to the lower H_2_S corrosion resistance of GBF. The microstructure of CGHAZ was mainly PF and GB, as well as some LB and M/A constituents. The lowest number of high angle grain boundaries and severe coarsening of grain size significantly reduced the crack propagation resistance for CGHAZ. The microstructure of the SCHAZ was homogeneous ferrite grains maintaining the rolling direction of the BM compared with the WM and CGHAZ, making SCHAZ exhibit better resistance to H_2_S corrosion and crack propagation.


## Figures and Tables

**Figure 1 materials-15-04458-f001:**
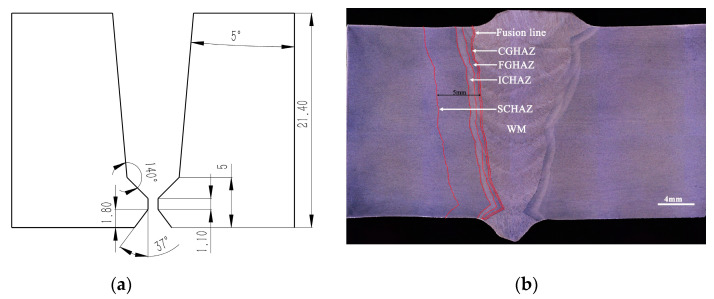
Macroscopic appearance of the X80 pipeline steel-welded joint. (**a**) welding groove structure; (**b**) cross-section of the weld joint.

**Figure 2 materials-15-04458-f002:**
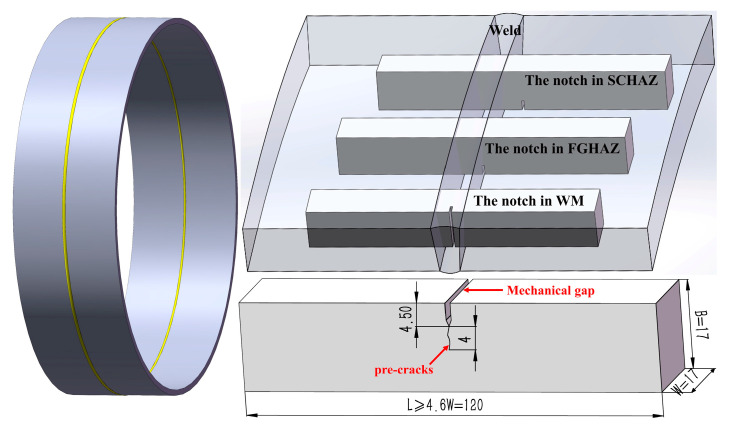
The locations of CTOD samples.

**Figure 3 materials-15-04458-f003:**
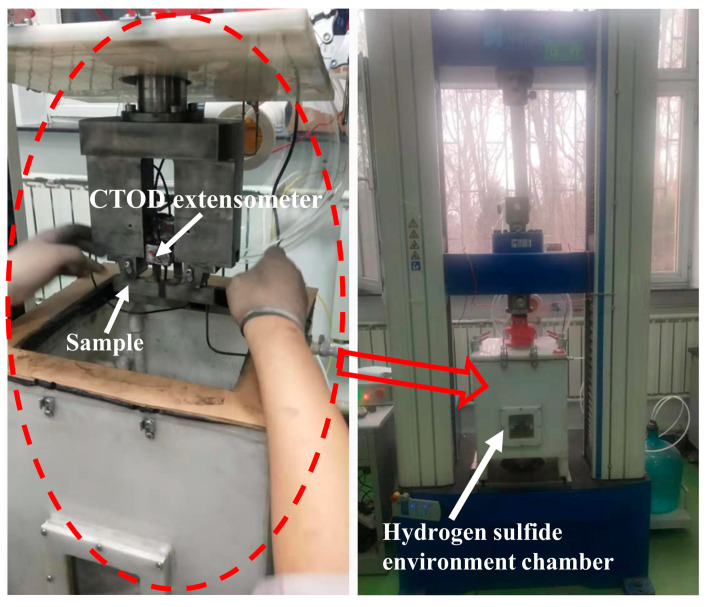
The equipment of hydrogen sulfide environment CTOD test.

**Figure 4 materials-15-04458-f004:**
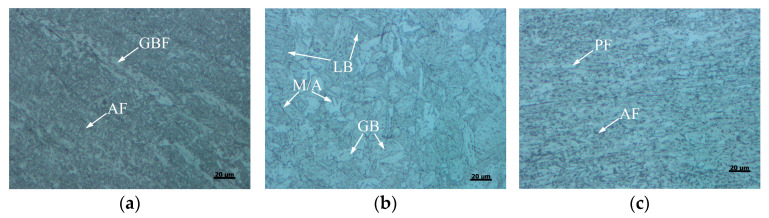
OM of the welded joint at different locations: (**a**) WM, (**b**) CGHAZ, (**c**) SCHAZ; SEM of the welded joint at different locations: (**d**) WM, (**e**) CGHAZ, (**f**) SCHAZ.

**Figure 5 materials-15-04458-f005:**
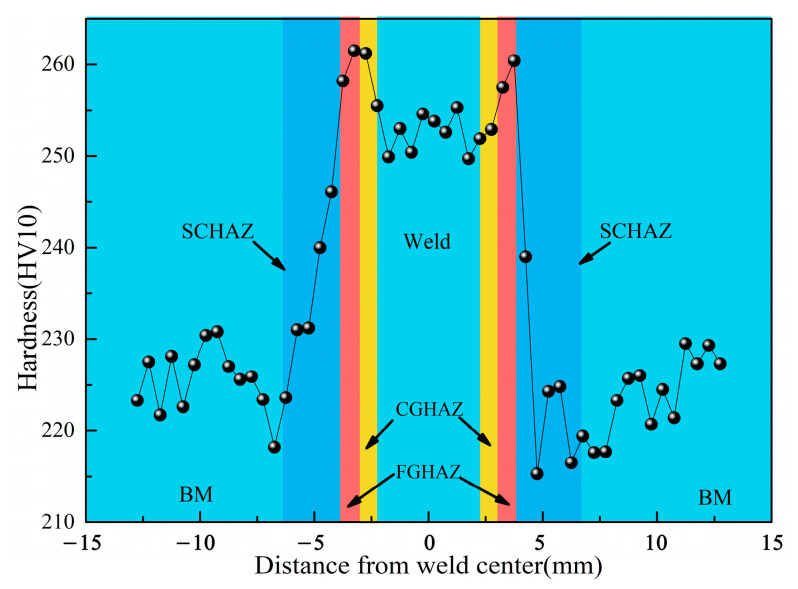
Hardness distribution of the X80 welded joint.

**Figure 6 materials-15-04458-f006:**
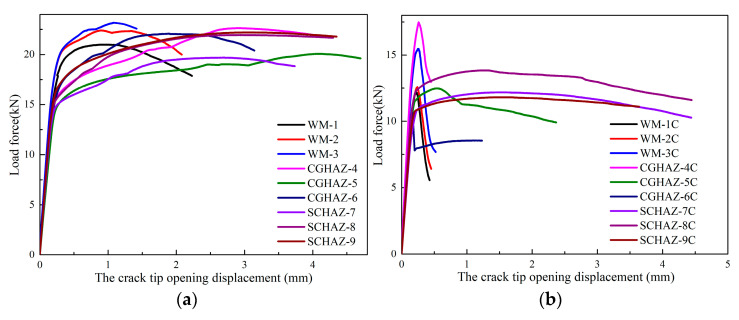
The loading curve of the CTOD test: (**a**) air environment, (**b**) hydrogen sulfide medium.

**Figure 7 materials-15-04458-f007:**
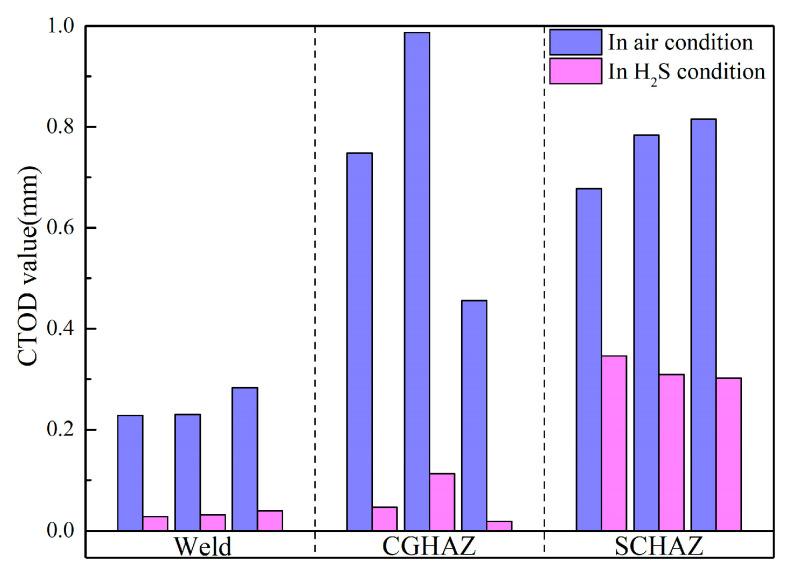
CTOD characteristic values.

**Figure 8 materials-15-04458-f008:**
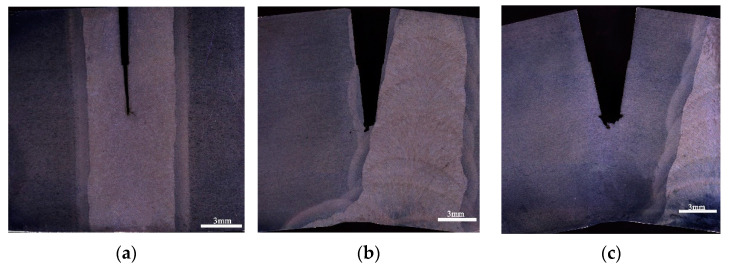
Typical CTOD profiles in the H_2_S-containing environment: (**a**) WM, (**b**) CGHAZ, (**c**) SCHAZ.

**Figure 9 materials-15-04458-f009:**
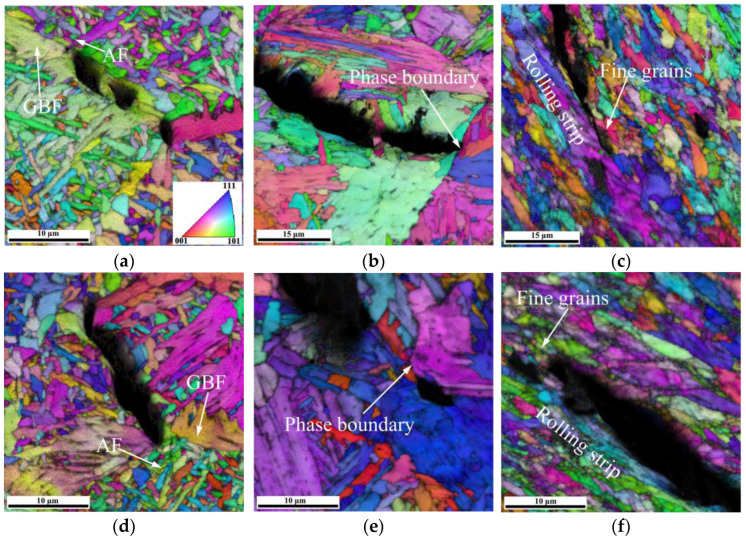
EBSD maps of secondary cracks in the H_2_S-containing environment: (**a**) WM, (**b**) CGHAZ, (**c**) SCHAZ; EBSD maps of secondary cracks in air environment: (**d**) WM, (**e**) CGHAZ, (**f**) SCHAZ.

**Figure 10 materials-15-04458-f010:**
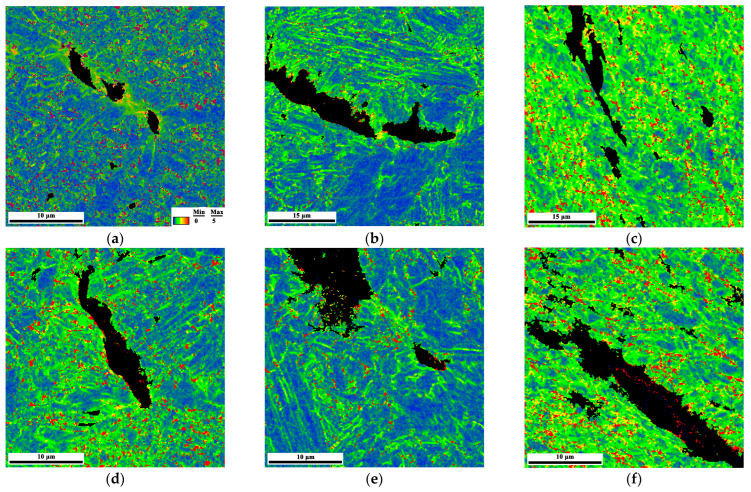
KAM maps of secondary cracks in the H_2_S-containing environment: (**a**) WM, (**b**) CGHAZ, (**c**) SCHAZ; KAM maps of secondary cracks in air environment: (**d**) WM, (**e**) CGHAZ, (**f**) SCHAZ.

**Figure 11 materials-15-04458-f011:**
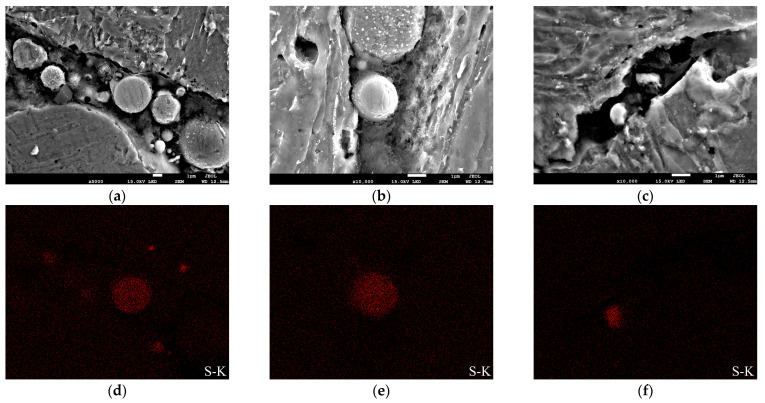
SEM morphology near cracks in the H_2_S-containing environment: (**a**) WM, (**b**) CGHAZ, (**c**) SCHAZ; S distributions analyzed via EDS mapping near cracks in the H_2_S-containing environment: (**d**) WM, (**e**) CGHAZ, (**f**) SCHAZ.

**Figure 12 materials-15-04458-f012:**
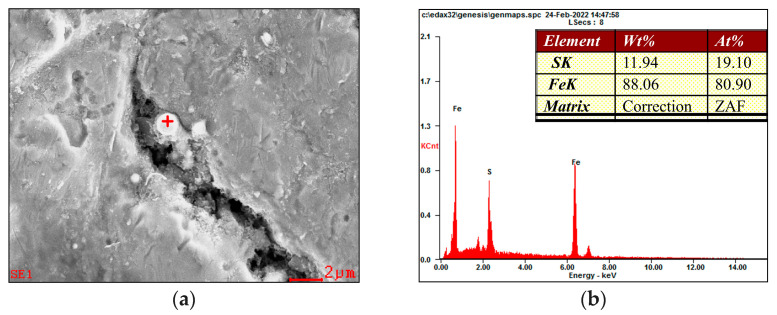
EDS analysis near the cracks: (**a**) SEM image; (**b**) element content (Test location as the red ‘+’ shown in Figure 12a).

**Figure 13 materials-15-04458-f013:**
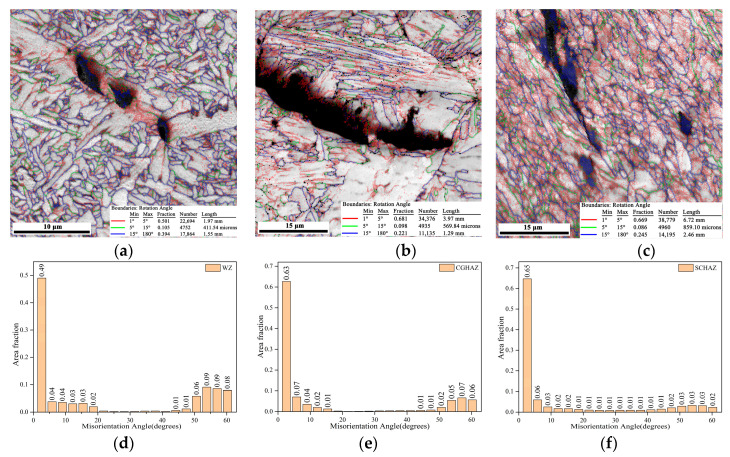
The distributions of grain orientation angles in different locations of the welded joint: (**a**,**d**) WM; (**b**,**e**) CGHAZ; (**c**,**f**) SCHAZ.

**Figure 14 materials-15-04458-f014:**
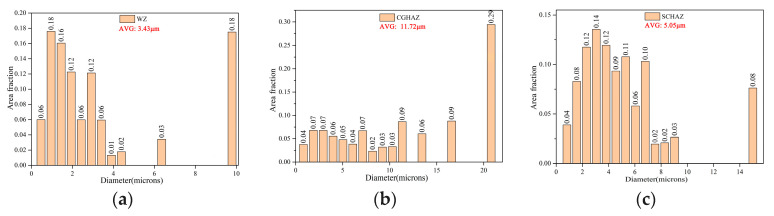
The distributions of grain sizes in different locations of the welded joint: (**a**) WM; (**b**) CGHAZ; (**c**) SCHAZ.

## Data Availability

Not applicable.

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
