# Peer review of "Effect of H2S Corrosion on the Fracture Toughness of the X80 Pipeline Steel Welded Joint"

_materials, 2022, doi:10.3390/ma15134458_

Round 1
Reviewer 1 Report
The manuscript presents an experimental investigation around the fracture toughness of X80 pipeline steel weld joints in both air and saturated H2S solution.
The experimental design is clear and the results are solid and could be valuable to design high-grade pipeline steel welded joints.
However, it should be noted that the current manuscript is more a technical report rather than a scientific paper. Nevertheless, I consider that the manuscript contributes to knowledge in the field and the subject has industrial relevance. For that reason, the paper has potential to be accepted for publication in this journal after some minor revision.
I recommend the authors to introduce more information and images about the welding parameters and the equipment that was used to weld the joints.
Please introduce pictures of the CTOD samples and the dimensions of the pre-cracks.
Provide more information about the calculations of the CTOD values.
There is a lack of information and figures about the equipment and experimental procedure used to simulate H2S environment.
Can the authors please comment if the conclusions from this study could be extrapolated for joints using other tube’s material?
Reviewer 2 Report
This manuscript deals with the experimental examination of the fracture toughness of X80 pipeline steel weld joints in saturated H2S solution. The evolution of microstructure in selected zones of the joints was analyzed.
In general, the manuscript is suitable for publication because the subject is relevant for the pipeline technology and in-service safety standards. Also, the experimental description denotes a very good level of expertise in the subject
However, from reading the manuscript many uncertainties arise that needs some clarification. In general, many unproven conclusions in the text and based on unconvincing arguments.
Introduction.
The introduction needs some organization to improve its readability.
I suggest starting (before line 38) with a brief description of the HAZ. Is Fig 1 a typical appearance for welded joints observed in different steels? Does HAZ controls the safety the in-service safety of a pipeline?
Also, there are an unclear sentence,
· Between lines 30-40
“ It is known that the CGHAZ near the fusion line (FL) would experience a longer period of high-temperature welding thermal cycling (about 1200 ℃) in the welding process, causing excessive growth of grain size [8]”
It seems that this sentence is an interpretation of ref [8] fragment. Please review and clarify the sentence.
At the end of the introduction a more complete experimental procedure is suggested (i.e. Brief sample description and experimental testing methods used).
Experimental methods
Line 94. How H2S was dosed?
Results and discussion.
Sections 3.1 to 3.4
These sections are difficult to read because the interpretation of morphological pictures, is not obvious (perhaps an expert eye for identifying visual characters is needed, or in-deep knowledge of the experimental methods)
· i.e., Fig. 8
Lines 161-162:” The cracks at the WM are inclined to propagate within the coarse GBF and be arrested on AF for both testing environments” Concerning this sentence, for an inexperienced reader the doubt is: Does colors in Fig 8 represent different phases?. Which color is ferrite?.
Section 3.5
Resistance to H2S. Since no electrochemical measurements were made, all interpretation is based on former’s work results. Many sentences are unclear i.e.
· Lines 255-259. Arguments from previous investigations were used to conclude that: “The combined effect of these factors resulted in CGHAZ exhibiting the weakest resistance to H2S corrosion “. This conclusion seems unconvincing.
Round 2
Reviewer 2 Report
I think that the authors responses are satisfactory and I do not have more remarks to add.
I my opinion this manuscript should be published